# Risk–Benefit Balance of Renin–Angiotensin–Aldosterone Inhibitor Cessation in Heart Failure Patients with Hyperkalemia

**DOI:** 10.3390/jcm11195828

**Published:** 2022-09-30

**Authors:** Shun Kohsaka, Suguru Okami, Naru Morita, Toshitaka Yajima

**Affiliations:** 1Department of Cardiology, Keio University School of Medicine, Tokyo 160-8582, Japan; sk@keio.jp; 2Cardiovascular, Renal, and Metabolism, Medical Affairs, AstraZeneca K.K., Osaka 530-0011, Japan; wolfsuguru@gmail.com (S.O.); naru.morita@astrazeneca.com (N.M.)

**Keywords:** heart failure, hyperkalemia, mineralocorticoid receptor antagonists, renin–angiotensin–aldosterone system inhibitors, risk–benefit

## Abstract

Background: Whether to continue renin–angiotensin–aldosterone system inhibitor (RAASi) therapy in patients with hyperkalemia remains a clinical challenge, particularly in patients with heart failure (HF), where RAASis remain the cornerstone of treatment. We investigated the incidence of dose reduction or the cessation of RAASis and evaluated the threshold of serum potassium at which cessation alters the risk–benefit balance. Methods: This retrospective analysis of a Japanese nationwide claims database investigated treatment patterns of RAASis over 12 months after the initial hyperkalemic episode. The incidences of the clinical outcomes of patients with RAASi (all ACEi/ARB/MRA) or MRA-only cessation (vs. non-cessation) were compared via propensity score-matched patients. A cubic spline regression analysis assessed the hazard of death resulting from treatment cessation vs. no cessation at each potassium level. Results: A total of 5059 hyperkalemic HF patients were identified; most received low to moderate doses of ACEis and ARBs (86.9% and 71.5%, respectively) and low doses of MRAs (76.2%). The RAASi and MRA cessation rates were 34.7% and 52.8% at 1 year post-diagnosis, while the dose reduction rates were 8.4% and 6.5%, respectively. During the mean follow-up of 2.8 years, patients who ceased RAASi or MRA therapies were at higher risk for adverse outcomes; cubic spline analysis found that serum potassium levels of <5.9 and <5.7 mmol/L conferred an increased mortality risk for RAASi and MRA cessation, respectively. Conclusions: Treatment cessation/dose reduction of RAASis are common among HF patients. The risks of RAASi/MRA cessation may outweigh the benefits in patients with mild to moderate hyperkalemia.

## 1. Introduction

Heart failure (HF) is a common condition with substantial morbidity and mortality worldwide. Major treatment advances have been made in recent years, and renin–angiotensin–aldosterone system inhibitors (RAASis), including angiotensin-converting enzyme inhibitors (ACEis), angiotensin II receptor blockers (ARBs), and mineralocorticoid receptor antagonists (MRAs), are guideline-recommended medications for patients with HF that reduce the risk of cardiovascular complications and improve survival. [1,2,3] Current clinical practice guidelines typically recommend the use of RAASis as foundations of pharmacotherapy for patients with HFrEF (class I, evidence level A) and partially recommend it for HFpEF and HFmEF patients (Class 2b, evidence level C). The up-titration RAASi medications to the maximum-tolerated doses is also recommended to improve clinical outcomes in HF patients. [4,5] Indeed, a recent meta-analysis of 14 randomized clinical trials investigating patients with HF and reduced ejection fraction (HFrEF) reported that higher doses of ACEis or ARBs significantly reduced the risk of HF-related hospitalizations compared to lower doses [6].

However, concerns about the adverse effects of RAASi therapy, primarily hyperkalemia, limit its use [7,8,9]. An increased risk of adverse clinical outcomes associated with hyperkalemia such as arrhythmia, myocardial infarction (MI), stroke, and all-cause mortality, particularly in patients with underlying kidney disease, has been reported [7,10,11,12,13,14,15] Accordingly, hyperkalemia often impedes the achievement of maximum RAASi dosing or results in dose reduction or cessation in HF patients [16,17].

While RAASi cessation or dose reduction is a practical option to lower serum potassium, it is also important to consider the risk–benefit balance in each HF patient individually. Although the benefits of RAASis for clinically important outcomes are well established, the clinical significance of cessation or dose reduction are poorly understood. In the present study, we investigated the ACEi/ARB/MRA treatment dose at baseline and dose reduction or cessation in hyperkalemic HF patients, and we assessed the subsequent effect of RAASi cessation on clinical outcomes. In addition, we assessed the risk–benefit balance of RAASi (all ACEi/ARB/MRA) or MRA (MRA only) cessation based on serum potassium levels.

## 2. Methods

### 2.1. Data Source

This study used a Japanese hospital claims database curated and maintained by Medical Data Vision Co., Ltd. (Tokyo, Japan). The database contains both administrative claims and laboratory data and is linked to the Diagnostic Procedure Combination (flat-fee payment system) inpatient hospital payment system, covering over 10 million patients across the country and corresponding to about 20% of the total population in Japan. Diagnoses of diseases were coded according to the International Classification of Diseases, 10th revision (ICD-10) [18]. Data collected between 1 April 2008 and 30 September 2018 were used in this study, including more than 25 million individual patient records from >380 hospitals across Japan.

The use of deidentified data was performed in compliance with local regulations. The study protocol was approved by the Clinical Research Promotion Network Japan, an independent ethics committee (protocol no. 2440023). Informed consent was not required, as patient records were anonymized and deidentified prior to access.

The authors had full access to all of the data in the study and take responsibility for its integrity and the data analysis. The data underlying the findings described in this manuscript cannot be shared publicly for contractual reasons; however, the data may be obtained in accordance with AstraZeneca’s data sharing policy, which is described at http://astrazenecagrouptrials.pharmacm.com/ST/Submission/Disclosure (accessed on 1 August 2022).

### 2.2. Study Design and Patient Selection

This was an observational, retrospective cohort study of HF patients with hyperkalemia. The following ICD-10 codes were used to identify patients with HF. I50.0: congestive heart failure, congestive heart disease, or right ventricular failure; I50.1: left ventricular failure, cardiac asthma, left heart failure, pulmonary edema, heart disease, or heart failure; and I50.9: heart failure (details unknown), heart failure, or myocardial insufficiency (not otherwise specified). The ICD-10 codes used to identify other high-risk comorbidities associated with hyperkalemia [13], including chronic kidney disease (CKD), diabetes mellitus, and hypertension, are listed in Appendix A. Other comorbidities associated with hyperkalemia are listed in Appendix A.

Patients with HF were included in the present analysis if they had experienced at least two hyperkalemic episodes (serum potassium ≥5.1 mmol/L) [13] within a 12-month period and were ≥45 years of age and received RAASi treatment at the time of the first hyperkalemic episode (the index date). Patients were excluded if they could not be followed for at least 12 months, were on dialysis prior to the index date, or had a cancer diagnosis. Then, patients were followed until the date of in-hospital death, the end of the study period (30 September 2018), or until emigration from the dataset (i.e., treatment cessation or transfer to a healthcare facility that did not provide hospital records to the database), whichever came first.

Information on treatments for hyperkalemia, including diuretics (thiazide and loop diuretics), glucose injection, calcium gluconate, sodium bicarbonate, and potassium binders, was collected. We also collected information regarding drugs associated with hyperkalemia, including RAASi drugs (i.e., ACEi, ARB, and MRA agents), azole antifungals, β-blockers, calcium channel blockers, cyclosporin, digoxin, heparin, non-steroidal anti-inflammatory drugs, potassium supplements, tacrolimus, trimethoprim, and systemic corticosteroids. Drug treatment records were collected for the 120 days prior to the index date; timing was based on commonly used intervals of drug prescription in clinical practice in Japan.

We assessed the frequency of RAASi cessation and dose reduction within the 12 months following the first hyperkalemic episode. RAASi cessation was defined as having no prescription of any RAASi for ≥30 days following the last day of any prior RAASi prescription. RAASi dose reduction was defined as switching to a lower dose of any RAASi among the high-, medium-, and low-dosage categories determined according to domestic prescription information; detailed information on dosage categories (high/medium/low) can be found in Appendix A. MRA cessation was defined as having no prescription for any MRA for ≥30 days following the last day of the prior MRA prescription (regardless of concomitant RAASi use).

### 2.3. Clinical Outcomes

The outcomes of interest included in-hospital death; hospitalization for MI, arrhythmia, or cardiac arrest; hospitalization for HF; and introduction of renal replacement therapy; definitions are listed in Appendix A. Mortality statistics of the general population in Japan indicate that in-hospital death represents the majority of fatal events [19].

### 2.4. Statistical Analysis

Continuous variables are reported as the mean, standard deviation (SD), and median; categorical variables are reported as the frequency and percent. Cumulative incidence for the first occurrence of hospitalization and hyperkalemic episode recurrence was estimated using the cumulative incidence function with death as a competing risk. The cumulative incidence of clinical outcomes was estimated using the Kaplan–Meier method. Missing data were not imputed.

Patients were categorized according to whether RAASi cessation occurred within 12 months after the index date. Propensity score (PS) matching was used to further adjust for imbalances in patient characteristics between the two groups using the covariates listed in Appendix A. To assess the validity of PS matching, standardized differences of patient characteristics were evaluated. A standardized difference >10% was considered to indicate a significant imbalance between two groups.

To assess the risk of RAASi cessation compared to no cessation, hazard ratios (HRs) of in-hospital death for RAASi cessation vs. no cessation were evaluated based on the potassium level at the index date. More specifically, the patients were stratified at intervals of 0.2 mmol/L according to their serum potassium level at the index date. The HRs of in-hospital death associated with RAASi cessation compared to no cessation were calculated based on the stratified potassium levels. Furthermore, cubic spline regression was used to analyze the relationship between the HRs and serum potassium levels for in-hospital death.

The analyses were performed in patients prescribed any RAASi medication and separately for the subset of patients prescribed MRAs. A *p*-value of <0.05 was considered statistically significant. All statistical analyses were performed using SAS version 9.4 (SAS Institute, Cary, NC, USA).

## 3. Results

### 3.1. Patients and Baseline Characteristics

Of the 1,208,894 patients screened, 5059 met both the inclusion and exclusion criteria and were included in this study (Figure 1). Within 12 months following the first hyperkalemic episode, RAASi or MRA cessation occurred in 1757 (34.7%) and 1172 (52.8%) patients, respectively. For the RAASi-treated patients, the mean age, follow-up period, and serum potassium level were 76.6 years, 2.8 years, and 5.4 mmol/L, respectively (Table 1). Most patients (4111 (81.3%)) received treatment with an ACEi and/or ARB, and 2220 patients (43.9%) had received an MRA, 66.4% of whom were treated with concomitant ACEi and/or ARB. The majority of the RAASi-treated patients (70.8%) were also diagnosed with CKD. Diabetes mellitus and hypertension were reported in 2505 (49.5%) and 4573 (90.4%) patients, respectively.

### 3.2. RAASi Treatment Patterns, Dose Reduction, and Cessation

Among all of the RAASi-treated patients (*n* = 5059), ARB was the most frequently used at baseline, with 2128 patients (42.1%), followed by MRA with 948 patients (18.7%); ARB + MRA, 772 (15.3%); and ACEi only with 610 patients (12.1%) (Table 2). At 12 months after the index date, cumulative cessation of RAASi treatment was observed in 1757/5059 (34.7%) patients (Figure 2, Table 2). As for RAASi treatment, MRA treatments were discontinued in 1172/2220 (52.8%) patients. The cessation rates according to drug combinations are shown in Appendix A.

The doses of ACEi, ARB, and MRA were examined according to the dosage category at the index date (Figure 3). Most ACEi or ARB prescriptions were doses in the medium or low category (ACEi: medium, 44.7%; low, 42.2%; ARB: medium, 49.4%; low, 22.1%), while most MRA prescriptions (76.2%) were low doses. RAASi dose reduction was reported in 427/5059 (8.4%) patients 1 year after the index date. Of those, 264/427 (62.8%) changed from the medium- to the low-dose category, and 130/427 (30.4%) changed from the high- to the medium-dose category. MRA dose reduction was reported in 144/2220 (6.5%) patients, of whom 113/144 (78.5%) and 22/144 (15.3%) were changed from the medium- to the low-dose and from the high- to the medium-dose categories, respectively (Appendix A).

### 3.3. Impact of RAASi Cessation on Clinical Outcomes

The clinical outcomes of patients with RAASi cessation were evaluated by comparing them to the PS-matched patients without RAASi cessation. The patients with RAASi cessation were at a higher risk of emergency room visits (HR: 1.62 [95% CI: 1.44–1.83]; *p* < 0.001), hospitalization (1.83 [1.68–2.00]; *p* < 0.001), hospitalization due to HF (1.72 [1.48–2.00]; *p* < 0.001), renal replacement therapy (1.72 [1.33–2.22]; *p* < 0.001), and hospitalization for MI, arrhythmia, or cardiac arrest (2.33 [1.70–3.18]; *p* < 0.001) compared to those without RAASi cessation (Figure 4). In the comparison between patients with and without MRA cessation, the patients with MRA cessation had a significantly higher risk of in-hospital death (HR: 1.21 [95% CI: 1.01–1.44]; *p* = 0.038), emergency room visits (1.35 [1.14–1.58]; *p* < 0.001), hospitalization (1.67 [1.48–1.89]; *p* < 0.001), hospitalization due to HF (1.75 [1.44–2.12]; *p* < 0.001), renal replacement therapy (2.08 [1.22–3.54]; *p* = 0.007), and hospitalization for MI, arrhythmia, or cardiac arrest (2.11 [1.40–3.18]; *p* < 0.001) compared to those without MRA cessation (Figure 5).

### 3.4. Cubic Spline Regression Analysis

To assess the hazards of RAASi or MRA cessation according to the baseline level of serum potassium, cubic spline regression analysis was performed. We employed a PS-matching analysis to compare the hazard of in-hospital death between patients with and without RAASi/MRA cessation (Figure 6). The risk of in-hospital death associated with RAASi or MRA cessation was found to outweigh that associated with no RAASi or MRA cessation when serum potassium levels were below 5.92 mmol/L or 5.71 mmol/L, respectively.

## 4. Discussion

In this retrospective cohort study, we investigated whether RAASi treatment cessation due to hyperkalemia changes the risk–benefit balance for adverse clinical events in patients with HF. We first examined RAASi treatment patterns, including dose reduction and cessation, in hyperkalemic HF patients and found suboptimal RAASi therapy with a high rate of cessation. Furthermore, RAASi or MRA cessation was associated with various adverse clinical outcomes when compared to PS-matched patients without RAASi or MRA cessation. In addition, our findings suggested that the risks of RAASi or MRA cessation would outweigh the benefits in HF patients with mild to moderate hyperkalemia.

Previously, in the UK, 22.9% of new RAASi users experienced cessation after the first hyperkalemic episode [20]. Similarly, a healthcare records study in the US reported that 16%–21% and 22%–27% of patients reduced or discontinued RAASi use after a hyperkalemia event, respectively [16]. A Swedish observational study also reported that in patients who developed hyperkalemia, a high proportion (47%) of patients discontinued MRA treatment, and dose reduction rates were 10% [21]. In that study, most patients (76%) who had discontinued MRA treatment were not reinitiated on therapy during the subsequent year. Although these studies did not focus only on hyperkalemic HF patients, our study also demonstrated that the cessation rates of RAASis or MRAs in HF patients at 1 year were 34.7% and 52.8%, respectively. Taken together, there seems to be a substantial gap between HF treatment guideline recommendations and the real-world implementation of RAASi or MRA therapy in hyperkalemic HF patients internationally.

We also found that patients who discontinued RAASi therapies had a significantly higher risk of hospitalization and a trend towards a higher risk of in-hospital mortality compared to those who maintained RAASi therapy. These findings are similar to those from previous reports showing that HF patients with hyperkalemia who either discontinued or received a suboptimal dose of a RAASi had a significantly higher risk of adverse clinical outcomes [16,17,22,23,24,25]. A previously published retrospective study also found a higher incidence of major adverse cardiac events and mortality in HF patients receiving <50% of the recommended RAASi dose compared to those receiving ≥50% of the recommended dose [17].

Current guidelines recommend short-term RAASi cessation or dose reduction for HF patients who experience hyperkalemia [4,5,26] or encourage use with caution in patients with elevated serum potassium [27]. On the other hand, suboptimal dosing and treatment cessation practices have been questioned [9,16]. In a multinational study (Sweden, UK and US) evaluating the use and dosing of guideline-directed medical therapy in HF patients initiated on RAASis, the authors found consistent patterns of low up-titration and the early cessation of RAASis in the three countries with different health care and economies [28].

A clinical trial investigating the dose-dependent effects of spironolactone, an MRA, on left ventricular function and exercise tolerance in patients with chronic HF found that 12 months of treatment with spironolactone improved left ventricular volume and function in a dose-dependent manner, with the greatest benefit being observed in the 50-mg group [29]. Additionally, patients treated with 50 mg had improved exercise tolerance, suggesting that spironolactone imparts important benefits to chronic HF patients, particularly at higher doses, which highlights the importance of achieving and maintaining RAASi dosing for improved clinical outcomes. In a Japanese study of 3717 patients hospitalized for acute decompensated HF, the use of an MRA at discharge was associated with a lower risk of readmission for HF [30]. In that study, 45.1% of patients were receiving MRA treatment at discharge, 93.6% of whom received spironolactone, and the median daily dose was 25 mg. Similarly, we found that 43.9% of the patients were receiving MRA therapy at baseline, with 76.2% of the prescriptions being for low doses. These data suggest that in Japanese real-world clinical practice, the majority of HF patients receive suboptimal doses of MRAs, while Japanese and European Society of Cardiology (ESC) guidelines recommend administering the maximum-tolerated RAASi dose for HFrEF patients [5,31,32].

Despite an increased risk of hyperkalemia, patients with renal dysfunction were shown to benefit from MRA treatment in both the RALES and EMPHASIS-HF trials [33,34,35]. The RALES data showed that the benefits of spironolactone treatment were maintained in patients with moderate hyperkalemia. More specifically, the clinical outcomes with spironolactone were superior to those with placebo when serum potassium levels remained <6.0 mmol/L [34,36]. Similarly, our data showed an increased risk of mortality by RAASi or MRA cessation in HF patients at serum potassium levels <5.92 or <5.71 mmol/L, respectively. Notably in our study, we compared the impact of RAASi cessation with the suboptimal RAASi group, which may underestimate the impact of cessation. Therefore, when optimal RAASi treatments are standardized, the impact of RAASi cessation may be greater. Currently available observational data also suggest that maintaining serum potassium concentrations between 4.0 and 5.5 mmol/L is associated with better clinical outcomes [37]. These findings, including those of our study, suggest that RAASi cessation should be reconsidered in HF patients with mild to moderate hyperkalemia.

An alternative approach to RAASi cessation or dose reduction for hyperkalemic HF patients is using potassium binders in addition to RAASi therapy. ESC HF guidelines updated in 2021 recommend the initiation of a potassium-lowering agent to encourage the use of maximal therapy in patients with serum potassium levels from >5.0 to ≤6.5 mmol/L and who are not on a maximal RAASi dose; [5] newly developed potassium-binding agents for the treatment of hyperkalemia may help to facilitate optimal RAASi treatment and improve clinical outcomes [5,38,39,40]. However, current Japanese HF guidelines have no recommendations for RAASi facilitation in hyperkalemia patients [26,31].

Although we examined a large sample from a data set extracted from a nationwide claims registry that reflects real-world treatment practices, there are still limitations to be considered for this study. First, because the hospital claims data were not collected for specific research purposes, clinical information that may have allowed for better patient characterization, such as dietary restrictions implemented to control hyperkalemia or other details regarding nutrition or cardiac function, such as the ejection fraction (EF), could not be extracted from the database. Second, selection bias may have been introduced, as in clinical practice, serum potassium levels are generally only measured in patients who are at a higher risk of developing hyperkalemia. Third, there may have been residual imbalances for some covariates despite our efforts to adjust for background patient demographic and disease characteristics. Forth, we were not able to directly assess the effect of potential drug–drug interactions that may influence potassium levels or other clinical outcomes. Additionally, the selection of variables included in PS modelling can affect both the validity and the precision of the effect estimates [41]. As such, confounding factors may have influenced the study outcomes. Finally, as this was an observational study, the results should be interpreted carefully; reported associations should not be considered to be indicative of a causal relationship. Future studies using a prospective cohort design may allow for the more rigorous collection of clinical information and more controlled drug administration.

In conclusion, we assessed the risk–benefit balance of RAASi or MRA treatment cessation in Japanese HF patients with hyperkalemia. Although confirmatory studies are needed, our results indicate that caution should be exercised when managing hyperkalemia by RAASi or MRA treatment cessation in patients with HF, especially in those with mild to moderate hyperkalemia.

## Figures and Tables

**Figure 1 jcm-11-05828-f001:**
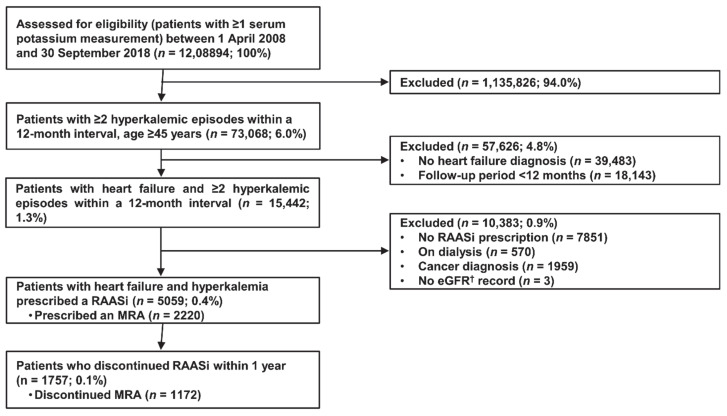
Flow diagram of patient inclusion in the study. Abbreviations: eGFR, estimated glomerular filtration rate; MRA, mineralocorticoid receptor antagonist; RAASi, renin–angiotensin–aldosterone system inhibitor.

**Figure 2 jcm-11-05828-f002:**
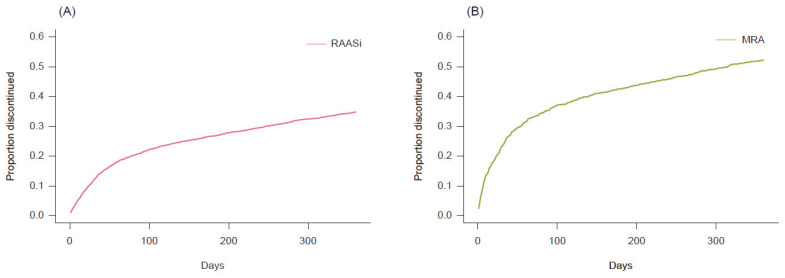
RAASi and MRA cessation rates within 12 months of the index date. (**A**) RAASi cessation (*n* = 5059). (**B**) MRA cessation (*n* = 2220). Abbreviations: MRA, mineralocorticoid receptor antagonist; RAASi, renin–angiotensin–aldosterone system inhibitor.

**Figure 3 jcm-11-05828-f003:**
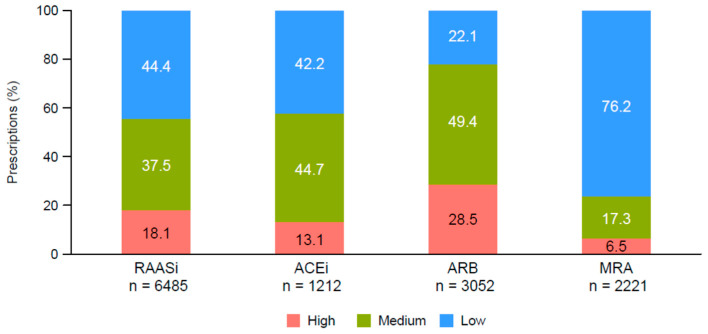
RAASi treatment according to dosage category (high, medium, or low) at the index date. Percentages were calculated using the total number of prescriptions. Abbreviations: ACEi, angiotensin-converting enzyme inhibitor; ARB, angiotensin receptor blocker; MRA, mineralocorticoid receptor antagonist; RAASi, renin–angiotensin–aldosterone system inhibitor.

**Figure 4 jcm-11-05828-f004:**
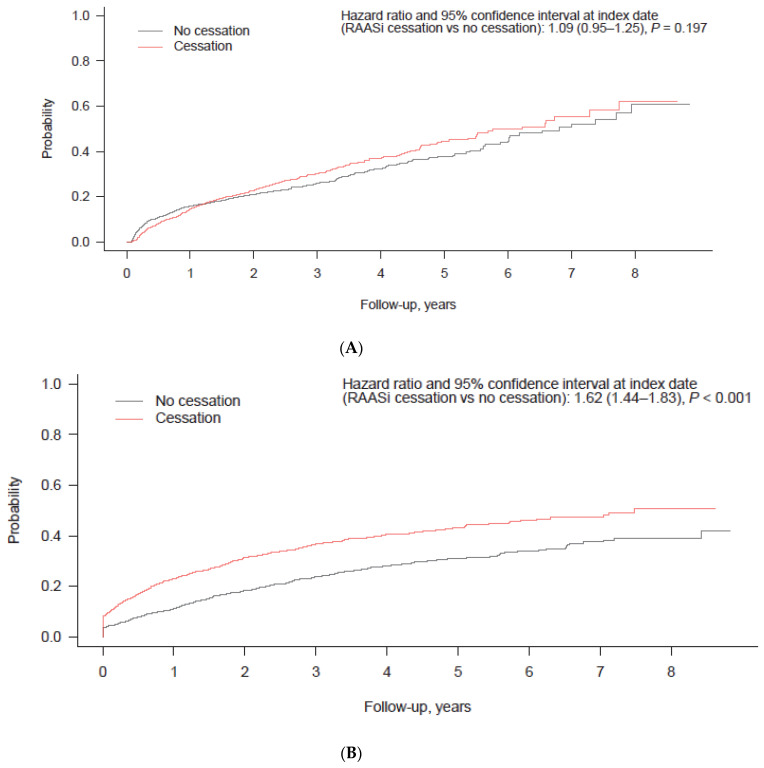
Cumulative incidence of clinical outcomes according to treatment cessation and non-cessation in propensity score-matched patients receiving any RAASi therapy (cumulative incidence function analysis). (**A**) In-hospital death; (**B**) emergency room visits; (**C**) hospitalization; (**D**) hospitalization due to heart failure; (**E**) renal replacement therapy; (**F**) hospitalization for MI, arrhythmia, or cardiac arrest. Abbreviations: MI, myocardial infarction; RAASi, renin–angiotensin–aldosterone system inhibitor.

**Figure 5 jcm-11-05828-f005:**
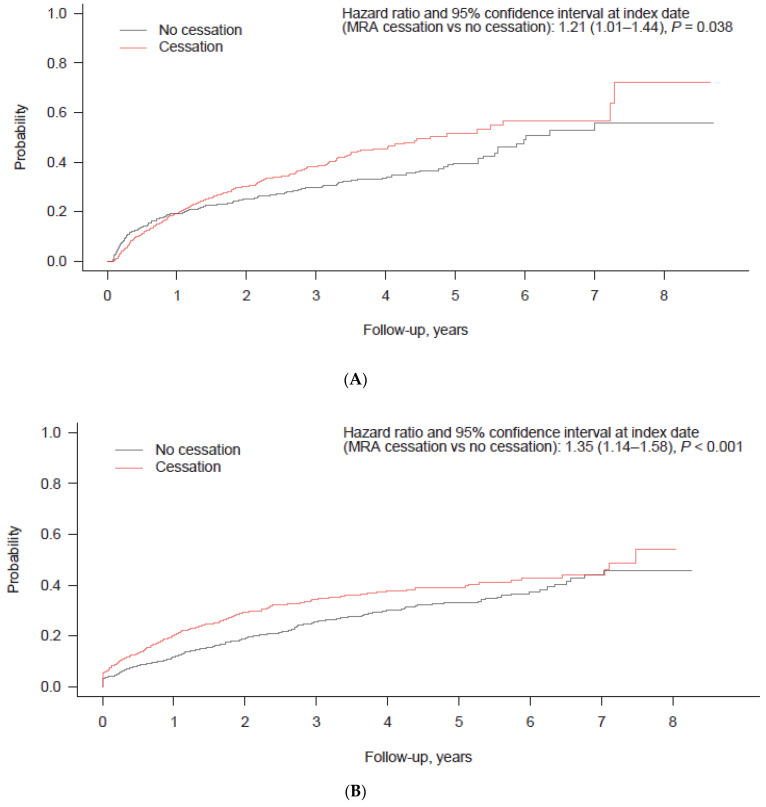
Cumulative incidence of clinical outcomes according to treatment cessation and non-cessation in propensity score-matched patients receiving MRA therapy (cumulative incidence function analysis). (**A**) In-hospital death; (**B**) emergency room visit; (**C**) hospitalization; (**D**) hospitalization due to heart failure; (**E**) renal replacement therapy; (**F**) hospitalization for MI, arrhythmia, or cardiac arrest. Abbreviation: MI, myocardial infarction; MRA, mineralocorticoid receptor antagonist.

**Figure 6 jcm-11-05828-f006:**
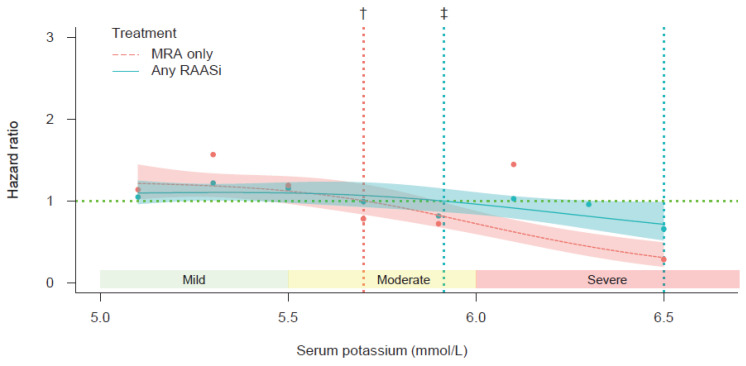
Cubic spline regression showing risk of in-hospital death as a function of serum potassium level. Data are plotted for propensity score-matched patients receiving any RAASi and for patients receiving an MRA only. The hazard ratio represents the risk of treatment cessation vs. no cessation. The dots in the figure show the point estimates of relative risks (i.e., hazard ratios of in-hospital death between patients with treatment cessation vs. no cessation) in each serum potassium value stratum. ^†^ Higher relative mortality risk of MRA cessation vs. no cessation up to serum potassium = 5.71 mmol/L. ^‡^ Higher relative mortality risk of RAASi cessation vs. no cessation up to serum potassium = 5.92 mmol/L. Abbreviations: MRA, mineralocorticoid receptor antagonist; RAASi, renin–angiotensin–aldosterone system inhibitor.

**Table 1 jcm-11-05828-t001:** Baseline patient characteristics.

	Any RAASi, Pre-Matching			Any RAASi, Post-Matching		MRA Group, Pre-Matching			MRA Group, Post-Matching	
	Any RAASi Therapy (*n* = 5059)	RAASi Cessation (*n* = 1757)	RAASi Non-Cessation (*n* = 3302)	RAASi Cessation (*n* = 1709)	RAASi Non-Cessation (*n* = 1709)	Total (*n* = 2220)	MRA Cessation (*n* = 1172)	MRA Non-Cessation (*n* = 1048)	MRA Cessation (*n* = 903)	MRA Non-Cessation (*n* = 903)
Age, years (mean ± SD)	76.63 ± 10.96	77.85 ± 10.98	75.98 ± 10.89	77.64 ± 10.99	77.78 ± 10.76	77.77 ± 10.95	78.19 ± 10.95	77.30 ± 10.93	77.70 ± 11.22	77.49 ± 10.89
Age group, years										
18–64	747 (14.8)	238 (13.5)	509 (15.4)	237(13.9)	202(11.8)	291 (13.1)	147 (12.5)	144 (13.7)	124 (13.7)	119 (13.2)
65–79	2019 (39.9)	622 (35.4)	1397 (42.3)	612 (35.8)	644 (37.7)	808 (36.4)	406 (34.6)	402 (38.4)	319 (35.3)	344 (38.1)
≥80	2293 (45.3)	897 (51.1)	1396 (42.3)	860 (50.3)	863 (50.5)	1121 (50.5)	619 (52.8)	502 (47.9)	460 (50.9)	440 (48.7)
Sex, male	2623 (51.8)	879 (50.0)	1744 (52.8)	865 (50.6)	841 (49.2)	1071 (48.2)	547 (46.7)	524 (50.0)	433 (48.0)	438 (48.5)
Follow-up period, years (mean ± SD)	2.8 ± 2.0	2.6 ± 1.7	2.9 ± 2.1	2.6 ± 1.8	2.6 ± 2.0	2.4 ± 1.9	2.5 ± 1.8	2.3 ± 2.1	2.4 ± 1.7	2.5 ± 2.1
Serum potassium at index date, mmol/L (mean ± SD)	5.37 ± 0.37	5.39 ± 0.40	5.36 ± 0.34	5.39 ± 0.41	5.36 ± 0.36	5.38 ± 0.39	5.39 ± 0.41	5.36 ± 0.35	5.39 ± 0.41	5.36 ± 0.36
Serum potassium group, mmol/L										
5.1–<5.5	3743 (74.0)	1257 (71.5)	2486 (75.3)	1227 (71.8)	1266 (74.1)	1637 (73.7)	845 (72.1)	792 (75.6)	656 (72.6)	686 (76.0)
5.5–<6.0	1020 (20.2)	382 (21.7)	638 (19.3)	365 (21.4)	342 (20.0)	455 (20.5)	256 (21.8)	199 (19.0)	194 (21.5)	170 (18.8)
6.0–<6.5	188 (3.7)	79 (4.5)	109 (3.3)	79 (4.6)	62 (3.6)	81 (3.6)	42 (3.6)	39 (3.7)	31 (3.4)	32 (3.5)
CKD	3584 (70.8)	1203 (68.5)	2381 (72.1)	1179 (69.0)	1175 (68.8)	1516 (68.3)	814 (69.5)	702 (67.0)	605 (67.0)	609 (67.4)
CKD stage *										
1	16 (0.4)	6 (0.5)	10(0.4)	6 (0.5)	5 (0.4)	5 (0.3)	2 (0.2)	3 (0.4)	2 (0.3)	2 (0.3)
2	173 (4.8)	70 (5.8)	103 (4.3)	68 (5.8)	65 (5.5)	72 (4.7)	44 (5.4)	28 (4.0)	27 (4.5)	24 (3.9)
3a	624 (17.4)	183 (15.2)	441 (18.5)	179 (15.2)	157 (13.4)	267 (17.6)	129 (15.8)	138 (19.7)	112 (18.5)	113 (18.6)
3b	1134 (31.6)	342 (28.4)	792 (33.3)	336 (28.5)	360 (30.6)	531 (35.0)	259 (31.8)	272 (38.7)	220 (36.4)	222 (36.5)
4	1196 (33.4)	431 (35.8)	765 (32.1)	421 (35.7)	413 (35.1)	536 (35.4)	320 (39.3)	216 (30.8)	200 (33.1)	207 (34.0)
5	441 (12.3)	171 (14.2)	270 (11.3)	169 (14.3)	175 (14.9)	105 (6.9)	60 (7.4)	45 (6.4)	44 (7.3)	41 (6.7)
Diabetes	2505 (49.5)	763 (43.4)	1742 (52.8)	755 (44.2)	743 (43.5)	950 (42.8)	489 (41.7)	461 (44.0)	397 (44.0)	398 (44.1)
Hypertension	4573 (90.4)	1499 (85.3)	3074 (93.1)	1489 (87.1)	1499 (87.7)	1897 (85.5)	980 (83.6)	917 (87.5)	771 (85.4)	785 (86.9)
Other comorbidities										
Myocardial infarction	458 (9.1)	169 (9.6)	289 (8.8)	165 (9.7)	140 (8.2)	205 (9.2)	112 (9.6)	93 (8.9)	90 (10.0)	77 (8.5)
Peripheral vascular disease	1153 (22.8)	364 (20.7)	789 (23.9)	361 (21.1)	352 (20.6)	481 (21.7)	227 (19.4)	254 (24.2)	188 (20.8)	200 (22.1)
Cerebrovascular disease	1547 (30.6)	503 (28.6)	1044 (31.6)	483 (28.3)	498 (29.1)	626 (28.2)	298 (25.4)	328 (31.3)	248 (27.5)	266 (29.5)
Chronic pulmonary disease	1208 (23.9)	400 (22.8)	808 (24.5)	395 (23.1)	383 (22.4)	569 (25.6)	274 (23.4)	295 (28.1)	219 (24.3)	232 (25.7)
Atrial fibrillation or atrial flutter	1708 (33.8)	593 (33.8)	1115 (33.8)	581 (34.0)	594 (34.8)	986 (44.4)	510 (43.5)	476 (45.4)	402 (44.5)	404 (44.7)
Valvular heart disease	1205 (23.8)	417 (23.7)	788 (23.9)	409 (23.9)	414 (24.2)	654 (29.5)	347 (29.6)	307 (29.3)	257 (28.5)	264 (29.2)
Acute kidney injury	201 (4.0)	76 (4.3)	125 (3.8)	74 (4.3)	73 (4.3)	89 (4.0)	43 (3.7)	46 (4.4)	36 (4.0)	37 (4.1)
Sepsis	733 (14.5)	242 (13.8)	491 (14.9)	239 (14.0)	235 (13.8)	342 (15.4)	149 (12.7)	193 (18.4)	129 (14.3)	136 (15.1)
Treatment for HF										
ACEi and/or ARB	4111 (81.3)	1399 (79.6)	2914 (88.2)	1376 (80.5)	1461 (85.5)	1474 (66.4)	800 (68.3)	674 (64.3)	610 (67.6)	592 (65.6)
Beta-blocker	2689 (53.2)	874 (49.7)	1815 (55.0)	855 (50.0)	944 (55.2)	1296 (58.4)	675 (57.6)	621 (59.3)	507 (56.1)	550 (60.9)
Digoxin	629 (12.4)	202 (11.5)	427 (12.9)	196 (11.5)	240 (14.0)	398 (17.9)	191 (16.3)	207 (19.8)	151 (16.7)	166 (18.4)
Inotrope	983 (19.4)	389 (22.1)	594 (18.0)	373 (21.8)	363 (21.2)	547 (24.6)	280 (23.9)	267 (25.5)	220 (24.4)	215 (23.8)
MRA	2220 (43.9)	949 (54.0)	1550 (46.9)	913 (53.4)	909 (53.2)	2220 (100.0)	1172 (100.0)	1048 (100.0)	903 (100.0)	903 (100.0)
Hyperkalemia treatment at index date										
Thiazide diuretics	208 (4.1)	55 (3.1)	153 (4.6)	55 (3.2)	76 (4.4)	80 (3.6)	34 (2.9)	40 (4.4)	31 (3.4)	39 (4.3)
Loop diuretics	2428 (48.0)	808 (46.0)	1620 (49.1)	791 (46.3)	800 (46.8)	1340 (60.4)	660 (56.3)	680 (64.9)	556 (61.6)	563 (62.3)
Glucose + insulin injection	90 (1.8)	30 (1.7)	60 (1.8)	29 (1.7)	40 (2.3)	46 (2.1)	23 (2.0)	23 (2.2)	18 (2.0)	18 (2.0)
Calcium gluconate	70 (1.4)	26 (1.5)	44 (1.3)	26 (1.5)	28 (1.6)	32 (1.4)	18 (1.5)	14 (1.3)	15 (1.7)	10 (1.1)
Sodium bicarbonate	527 (10.4)	211 (12.0)	316 (9.6)	209 (12.2)	153 (9.0)	201 (9.1)	109 (9.3)	92 (8.8)	95 (10.5)	74 (8.2)
Potassium binder (SPS/CPS)	352 (7.0)	129 (7.3)	223 (6.8)	125 (7.3)	118 (6.9)	91 (4.1)	63 (5.4)	28 (2.7)	50 (5.5)	26 (2.9)

Data are *n* (%) unless otherwise stated. * Percent calculated based on total patients diagnosed with CKD (*n* = 3584 for any RAASi and *n* = 1516 for MRA only). Abbreviations: ACEi, angiotensin-converting enzyme inhibitor; ARB, angiotensin receptor blocker; CKD, chronic kidney disease; CPS, calcium polystyrene sulfonate; HF, heart failure; MRA, mineralocorticoid receptor antagonist; RAASi, renin–angiotensin–aldosterone system inhibitor; SD, standard deviation, SPS, sodium polystyrene sulfonate.

**Table 2 jcm-11-05828-t002:** RAASi treatment patterns at baseline.

	Baseline		1 Year after Index Date
	*n* = 5059		*n* = 4175
Any RAASi	5059 (100)		2913 (69.8)
	**No cessation** 3302	**Cessation** 1757	
ACEi only	610 (12.1)		402 (9.6)
ARB only	2128 (42.1)		1537 (36.8)
ACEi + ARB	101 (2.0)		70 (1.7)
Any MRA	2220 (43.9)		907 (21.7)
	**No cessation** 1048	**Cessation** 1172	
MRA only	948 (18.7)		315 (7.5)
ACEi + MRA	464 (9.2)		198 (4.7)
ARB + MRA	772 (15.3)		377 (9.0)
ACEi + ARB + MRA	36 (0.7)		14 (0.3)

Data are *n* (%). Abbreviations: ACEi, angiotensin-converting enzyme inhibitor; ARB, angiotensin receptor blocker; MRA, mineralocorticoid receptor antagonist; RAASi, renin–angiotensin–aldosterone system inhibitor.

## Data Availability

The data included in this manuscript were used under contract with the supplier (Medical Data Vision Co., Ltd.) and cannot be freely distributed by the authors.

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
