# Peer review of "Risk–Benefit Balance of Renin–Angiotensin–Aldosterone Inhibitor Cessation in Heart Failure Patients with Hyperkalemia"

_jcm, 2022, doi:10.3390/jcm11195828_

Round 1
Reviewer 1 Report
Dear Author(s),
Thank you very much for this interesting, well-written, clinically relevant, and highly important manuscript. To the best of my knowledge, the body of literature regarding risk-benefit balance of RAASi cessation/dose-reduction in hyperkalemic HF patients is still relatively scarce. Thus, this topic is highly important. One of the cornerstone findings of this retrospective cohort study are the results of the subsequent cubic spline analysis by which you identified that serum potassium levels of <5.9 and <5.7 mmol/L conferred increased mortality risk for RAASi and MRA cessation, respectively.
Moreover, to deduce, methodology is adequate, results are interesting and clearly presented, most important study limitations are noted, and conclusions are supported by the results.
However, I have several minor suggestions in order to help you improve the quality of the manuscript even more:
(I) Introduction can be improved. Please implement grade of recommendation and level of evidence for pharmacological therapy options (especially for RAASi) for HF according to different types (HFrEF, HFmrEF, HFpEF). When providing this info please take into consideration both (newest) AHA, ESC and Japanese HF guidelines.
(II) Can you provide subanalysis data (this can be under secondary otucomes) according to HF types (HFrEF, HFmrEF, HFpEF). If not, please mention this issue in the limitation section. Do you believe that cut-off serum potassium levels, determined by spline cubic analysis, can significantly differ according to different HF types?
(III) Did you check for potential drug-drug interactions within groups that could have influenced potassium levels or some of the measured clinical outcomes? Please provide data or consider mentioning this in the limitation section.
(IV) After matching, the percentage of DM patients is congruent within groups. However, did you check SGLT2i (dapagliflozin, empagliflozin) prescription rates within groups? This is highly important, since SGLT2i demonstrated their efficacy for HF (especially rEF; but also in mrEF and pEF according to most recent findings from DELIVER trial). If you do not have that data, consider mentioning it in the limitation section since this may be a potential bias.
Best regards, Reviewer
Author Response
Sep 24th, 2022
Dr. Emmanuel Andrès
Editor-in-Chief
Dear Dr Andrès
Manuscript reference [JCM] Manuscript ID: jcm-1918447
Please find the revised version of our manuscript, entitled “Risk-Benefit Balance of Renin–Angiotensin–Aldosterone Inhibitor Cessation in Heart Failure Patients with Hyperkalemia”, in consideration for publication in Journal of Clinical Medicine.
We edited the manuscript carefully in line with the suggestions and comments made by the reviewers, and will enclose two versions of the manuscript, a tracked and clean copy in Word file.
Our responses to the reviewers’ comments are presented in the following pages where the original comments and our responses are described in italic and plain font, respectively. In addition, major modifications in the main text made according to each reviewer`s comment are shown in red in the respective response.
We believe the comments from the reviewers have significantly improved the quality of our manuscript, and we hope that the current revised manuscript with our accompanying responses have properly covered all the points raised by reviewers to be suitable for publication in Journal of Clinical Medicine.
Yours sincerely,
Toshitaka Yajima, M.D., Ph.D.
AstraZeneca K.K.
Tower B Grand Front Osaka, 3-1 Ofukacho, Kita-ku, Osaka-shi,
Osaka 530-0011, JAPAN
Email: Toshitaka.Yajima@astrazeneca.com
Tel.: +81-6-4802-3600
Comments from Reviewer #1
- Introduction can be improved. Please implement grade of recommendation and level of evidence for pharmacological therapy options (especially for RAASi) for HF according to different types (HFrEF, HFmrEF, HFpEF). When providing this info please take into consideration both (newest) AHA, ESC and Japanese HF guidelines.
Response: We thank the reviewer for careful suggestions. We have revised our manuscript based on his/her comments.
The references we used as AHA and ESC guideline in this manuscript are as below, and is the most updated HF clinical practice guideline from the US, Europe and Japan.
AHA: Heidenreich PA, Bozkurt B, Aguilar D, Allen LA, Byun JJ, Colvin MM, Deswal A, Drazner MH, Dunlay SM, Evers LR, et al. 2022 AHA/ACC/HFSA guideline for the management of heart failure: A report of the American College of Cardiology/American Heart Association Joint Committee on Clinical Practice Guidelines. J Am Coll Cardiol. 2022;79:e263–e421.
ESC: McDonagh TA, Metra M, Adamo M, Gardner RS, Baumbach A, Böhm M, Burri H, Butler J, Čelutkienė J, Chioncel O, et al. 2021 ESC Guidelines for the diagnosis and treatment of acute and chronic heart failure. Eur Heart J. 2021;42:3599–3726.
For Japanese clinical practice guideline, we originally used the full version that was published in 2017. Focused update of the guideline was provided in 2021, and we have updated the reference accordingly:
Before: Tsutsui H, Isobe M, Ito H, Ito H, Okumura K, Ono M, Kitakaze M, Kinugawa K, Kihara Y, Goto Y, et al. JCS 2017/JHFS 2017 Guideline on diagnosis and treatment of acute and chronic heart failure – Digest version. Circ J. 2019;83:2084–2184.
Updated: Tsutsui H, Ide T, Ito H, Kihara Y, Kinugawa K, Kinugawa S, Makaya M, et al. JCS/JHFS 2021 Guideline Focused Update on Diagnosis and Treatment of Acute and Chronic Heart Failure. J Card Fail. 2021 Dec;27(12):1404-1444.
RAASi has been recommended in the above guidelines as follows:
ESC : HFrEF (Class Ⅰ level A) and HFmrEF (ClassⅡb level C)
AHA : HFrEF (COR a LOE A), HFmrEF and Hfimp HF (COR 2b LOE B-NR), and
HFpEF (COR 2b LOE B-R)
JCS : HFrEF (Class Ⅰ level A), HFpEF (Class Ⅱb, Level C)
Incorporating above we have made following modifications in our revised manuscript.
Changes in the manuscript:
P3: Major treatment advances have been made in recent years, and the renin–angiotensin–aldosterone system inhibitors (RAASis), including angiotensin-converting enzyme inhibitors (ACEis), angiotensin II receptor blockers (ARBs), and mineralocorticoid receptor antagonists (MRAs) are guideline-recommended medications for patients with HF that reduce the risk of cardiovascular complications and improve survival.1-3 Current clinical practice guidelines typically recommend use of RAASi as foundations of pharmacotherapy for patients with HFrEF (class I, evidence level A) and partially recommend it for HFpEF and HFmEF patients (Class 2b, evidence level C). It is also recommended to up-titrate RAASi medications to the maximum tolerated doses to improve clinical outcomes in HF patients.4,5
Modulation of the renin-angiotensin-aldosterone (RAAS) and sympathetic nervous systems with angiotensin-converting enzyme inhibitors (ACE-I) or an angiotensin receptor-neprilysin inhibitor (ARNI), beta-blockers, and mineralocorticoid receptor antagonists (MRA) has been shown to improve survival, reduce the risk of HF hospitalizations, and reduce symptoms in patients with HFrEF. These drugs serve as the foundations of pharmacotherapy for patients with HFrEF. The triad of an ACE-I/ARNI, a beta-blocker, and an MRA is recommended as cornerstone therapies for these patients, unless the drugs are contraindicated or not tolerated.
- Can you provide subanalysis data (this can be under secondary otucomes) according to HF types (HFrEF, HFmrEF, HFpEF). If not, please mention this issue in the limitation section. Do you believe that cut-off serum potassium levels, determined by spline cubic analysis, can significantly differ according to different HF types?
Response: We thank the reviewer for pointing out this important view for HF. Though we understand EF data is important in the treatment of HF, unfortunately MDV database does not include EF information. Thus, we have added this point in limitation. We therefore, also do not have clear answer whether cut-off of serum potassium level differ by HF type. However, ‘LVEF’ was added as within the Japanese coding system this April (2022), We would like to keep this question in the future research.
Change in the manuscript:
P15: Although we examined a large sample from a data set extracted from a nationwide claims registry that reflects real-world treatment practices, there are still limitations to be considered for this study. First, because the hospital claims data were not collected for specific research purposes, clinical information that may have allowed for better patient characterization, such as dietary restrictions implemented to control hyperkalemia, other details regarding nutrition, or cardiac function such as ejection fraction (EF) could not be extracted from the database.
- Did you check for potential drug-drug interactions within groups that could have influenced potassium levels or some of the measured clinical outcomes? Please provide data or consider mentioning this in the limitation section.
Response: We thank the reviewer for the valuable comments. We did not check the drug-drug interactions within groups, so we have added this point in the limitation. Since we have adjusted the patient background including HK medications using PS matching method at baseline, we believe the influence of potential drug-drug interaction on differences in potassium levels or clinical outcomes should be minimized when comparing two groups. The HK medications used for PS matching are as follows;
|
Cessation (n=1709) |
No cessation(n=1709) |
|
|
||
|
N |
% |
N |
% |
p-value |
Std.diff |
Hyper K treatment at the index date (n, %) |
|
|
|
|
|
|
Thiazide diuretics |
55 |
3.22% |
76 |
4.45% |
0.0645 |
-0.064 |
Loop diuretics |
791 |
46.28% |
800 |
46.81% |
0.7505 |
-0.0106 |
Glucose injection+Insulin |
29 |
1.70% |
40 |
2.34% |
0.1724 |
-0.0458 |
Calcium gluconate |
26 |
1.52% |
28 |
1.64% |
0.7855 |
-0.0094 |
Sodium bicarbonate |
209 |
12.23% |
153 |
8.95% |
0.0018 |
0.1066 |
Potassium binder (SPS / CPS) |
125 |
7.31% |
118 |
6.90% |
0.6392 |
0.0159 |
Change in the manuscript:
P15: Third, there may have been residual imbalances for some covariates despite our efforts to adjust for background patient demographic and disease characteristics. Forth, we were not able to directly assess the effect of potential drug-drug interactions which may give influence in potassium levels or some clinical outcomes.
- After matching, the percentage of DM patients is congruent within groups. However, did you check SGLT2i (dapagliflozin, empagliflozin) prescription rates within groups? This is highly important, since SGLT2i demonstrated their efficacy for HF (especially rEF; but also in mrEF and pEF according to most recent findings from DELIVER trial). If you do not have that data, consider mentioning it in the limitation section since this may be a potential bias.
Response: We appreciate reviewers’ comment on SGLT2i since its use is becoming increasingly important in management of HF patients.
SGLT2i indication for HF was approved in 2020 in Japan, and our study used database between April 1, 2008 and September 30, 2018 where SGLT2i was used only in less than 1% of the subject (please see table below). Therefore, we believe that SGLT2i use had minimal or no impact on our study.
|
Cessation (n=1709) |
No cessation(n=1709) |
|
|
||
|
N |
% |
N |
% |
p-value |
Std.diff |
DPP4 |
299 |
17.50% |
284 |
16.62% |
0.4876 |
0.0233 |
SGLT2 |
8 |
0.47% |
13 |
0.76% |
0.2752 |
-0.0374 |
Insulin |
395 |
23.11% |
339 |
19.84% |
0.0169 |
0.0799 |
GLP-1 |
13 |
0.76% |
12 |
0.70% |
0.8415 |
0.0069 |
FDC_DM |
6 |
0.35% |
16 |
0.94% |
0.033 |
-0.0732 |
SU |
137 |
8.02% |
148 |
8.66% |
0.4875 |
-0.0233 |
Alpha-GI |
141 |
8.25% |
147 |
8.60% |
0.7087 |
-0.0126 |
Glitazone |
30 |
1.76% |
30 |
1.76% |
1 |
0 |
Glinide |
32 |
1.87% |
35 |
2.05% |
0.7098 |
-0.0127 |
BG |
83 |
4.86% |
106 |
6.20% |
0.0873 |
-0.0589 |

Reviewer 2 Report
This is an interesting study conducted on a large study popualtion
Some clarificationmay improve readability adn impact of the manuscript
1- ho wmany subjects with hyperkalyemia were assuming both RAAS and MRA?
2- was the proportrion/percentage different in subjects with or without HF and with or without hyperalyemia?
3- which dosage of MRA - alone or in combination with RAAS - was taken in subjects with HF and hyperkalyemia?
Was this dosage superior to the target dosage recently recommended for HF pateints (see, for instance, ESC recommendations)?
4- any insight about other characteristics of the subjects with hyperkalyemia?
Given the role of MR receptors on adipocytes, were subjects with hyperkalyemia and MRA treatment more likely "metabolically impaired" (see PMID: 31843490)?
Author Response
Sep 24th, 2022
Dr. Emmanuel Andrès
Editor-in-Chief
Dear Dr Andrès
Manuscript reference [JCM] Manuscript ID: jcm-1918447
Please find the revised version of our manuscript, entitled “Risk-Benefit Balance of Renin–Angiotensin–Aldosterone Inhibitor Cessation in Heart Failure Patients with Hyperkalemia”, in consideration for publication in Journal of Clinical Medicine.
We edited the manuscript carefully in line with the suggestions and comments made by the reviewers, and will enclose two versions of the manuscript, a tracked and clean copy in Word file.
Our responses to the reviewers’ comments are presented in the following pages where the original comments and our responses are described in italic and plain font, respectively. In addition, major modifications in the main text made according to each reviewer`s comment are shown in red in the respective response.
We believe the comments from the reviewers have significantly improved the quality of our manuscript, and we hope that the current revised manuscript with our accompanying responses have properly covered all the points raised by reviewers to be suitable for publication in Journal of Clinical Medicine.
Yours sincerely,
Toshitaka Yajima, M.D., Ph.D.
AstraZeneca K.K.
Tower B Grand Front Osaka, 3-1 Ofukacho, Kita-ku, Osaka-shi,
Osaka 530-0011, JAPAN
Email: Toshitaka.Yajima@astrazeneca.com
Tel.: +81-6-4802-3600
Comments from Reviewer #2
- how many subjects with hyperkalyemia were assuming both RAAS and MRA?
Response: As described in the below table, the rate of ARB or ACEi and MRA is described in Table 2. ACEi +MRA, ARB + MRA, and ACEi + ARB + MRA was prescribed to 464 (9.2%), 772 (15.3%), and 36 (0.7%), respectively.
|
Baseline |
|
|
N=5,059 |
|
Any RAASi |
5,059 (100) |
|
No cessation 3,302 |
Cessation 1,757 |
|
ACEi only |
610 (12.1) |
|
ARB only |
2,128 (42.1) |
|
ACEi + ARB |
101 (2.0) |
|
Any MRA |
2,220 (43.9) |
|
No cessation 1,048 |
Cessation 1,172 |
|
MRA only |
948 (18.7) |
|
ACEi + MRA |
464 (9.2) |
|
ARB + MRA |
772 (15.3) |
|
ACEi + ARB + MRA |
36 (0.7) |
- Was the proportion/percentage different in subjects with or without HF and with or without hyperalyemia?
Response: We thank the reviewer for the valuable comment. Though proportion of hyperkalemia in patients with or without HF is interesting topic, in the present study, we included HF patient with hyperkalemia and assessed the subsequent effect of RAASi cessation on clinical outcomes. So, we do not have data for patients without HF
- which dosage of MRA - alone or in combination with RAAS - was taken in subjects with HF and hyperkalyemia? Was this dosage superior to the target dosage recently recommended for HF pateints (see, for instance, ESC recommendations)?
Response: Thank you very much for your comment. In this study MRA is included in RAASi. So we do not have data dose information for MRA with RAASi.
The dose information is described in Fig 3 also in page 9 of the manuscript. Notably, most ACEi or ARB were prescribed in doses within the medium or low category (ACEi: medium, 44.7%; low, 42.2%; ARB: medium, 49.4%; low, 22.1%), while most MRA prescriptions (76.2%) were low doses. RAASi dose reduction was reported in 427/5,059 (8.4%) patients 1 year after the index date.
This dose analyses was based on Japanese guideline recommended maximum dose. The result of the analysis has been added as Table S3.
- Any insight about other characteristics of the subjects with hyperkaliemia?
Given the role of MR receptors on adipocytes, were subjects with hyperkaliemia and MRA treatment more likely "metabolically impaired" (see PMID: 31843490)?
Response: Thank you very much for introducing us an interesting article. It is interesting that MRAs have influence on reducing fat mass and improve glucose tolerance, and that novel MRAs may be a potential compound for subjects with metabolic disorders. However, our study is retrospective database study and has not analyze metabolically impaired. Thus it is difficult for us to give insights for this point.